# Investigation of Mitochondrial Adaptations to Modulation of Carbohydrate Supply during Adipogenesis of 3T3-L1 Cells by Targeted ^1^H-NMR Spectroscopy

**DOI:** 10.3390/biom11050662

**Published:** 2021-04-29

**Authors:** Manon Delcourt, Virginie Delsinne, Jean-Marie Colet, Anne-Emilie Declèves, Vanessa Tagliatti

**Affiliations:** 1Metabolic and Molecular Biochemistry Unit, Faculty of Medicine and Pharmacy, Research Institute for Health Sciences and Technology, UMONS, 20 Place du Parc, 7000 Mons, Belgium; anne-emilie.decleves@umons.ac.be; 2Human Biology and Toxicology Unit, Faculty of Medicine and Pharmacy, Research Institute for Health Sciences and Technology, UMONS, 20 Place du Parc, 7000 Mons, Belgium; virginie.delsinne@umons.ac.be (V.D.); jean-marie.colet@umons.ac.be (J.-M.C.); vanessa.tagliatti@umons.ac.be (V.T.)

**Keywords:** carbohydrates, mitochondria, adaptation, metabolism, metabolomics, adipogenesis

## Abstract

(1) Background: White adipose tissue (WAT) is a dynamic and plastic tissue showing high sensitivity to carbohydrate supply. In such a context, the WAT may accordingly modulate its mitochondrial metabolic activity. We previously demonstrated that a partial replacement of glucose by galactose in a culture medium of 3T3-L1 cells leads to a poorer adipogenic yield and improved global mitochondrial health. In the present study, we investigate key mitochondrial metabolic actors reflecting mitochondrial adaptation in response to different carbohydrate supplies. (2) Methods: The metabolome of 3T3-L1 cells was investigated during the differentiation process using different glucose/galactose ratios and by a targeted approach using ^1^H-NMR (Proton nuclear magnetic resonance) spectroscopy; (3) Results: Our findings indicate a reduction of adipogenic and metabolic overload markers under the low glucose/galactose condition. In addition, a remodeling of the mitochondrial function triggers the secretion of metabolites with signaling and systemic energetical homeostasis functions. Finally, this study also sheds light on a new way to consider the mitochondrial metabolic function by considering noncarbohydrates related pathways reflecting both healthier cellular and mitochondrial adaptation mechanisms; (4) Conclusions: Different carbohydrates supplies induce deep mitochondrial metabolic and function adaptations leading to overall adipocytes function and profile remodeling during the adipogenesis.

## 1. Introduction

Today, white adipose tissue (WAT) is no longer considered a simple storage container for excessive nutrients. As other organs showing extraordinary plastic features, WAT exhibits cellular mechanisms of adaptation essential to maintain the overall systemic energy and metabolic homeostasis. Through a mechanistic approach, it is accepted now that WAT dynamics are challenged by several environmental features in both physiological and pathological contexts, such as obesity and metabolic disorders. Those adaptations lead to cellular changes known as WAT remodeling, including processes leading to the expansion of the tissue storage capacity by inducing an increase of adipocytes number. In this context, adipogenesis, the mechanism through which precursors cells differentiate into mature and functional adipocytes, has been extensively studied [1,2]. Adipogenesis is highly sensitive to the cell metabolic status and finely regulated by master gene regulators [3,4,5]. The 3T3-L1 cell line has been extensively used to study adipogenesis and is now well characterized in terms of transcriptional and proteomic events occurring through the differentiation process [6,7]. The same is not true for metabolic changes in differentiating adipocytes. To orchestrate this complex metabolic landscape remodeling, mitochondria play a central role [8,9]. Indeed, as crucial metabolic sensors, mitochondria are found at the crossroad of several biosynthetic pathways, especially focused on ATP production [10,11]. In addition, mitochondria are increasingly considered signaling hubs [10,12,13,14].

To better understand the influence of carbohydrate supply, both in terms of quantity and quality, on WAT plasticity, previous studies focused on studying how nutrients could modulate adipogenesis associated with WAT expansion [15,16,17]. In this context, we previously reported that the partial replacement of glucose by galactose in the culture medium of 3T3-L1 cells had an anti-adipogenic effect and globally improved their mitochondrial health [18]. This study also underlined important changes regarding mitochondrial and metabolic enzyme expression, suggesting a deep impact on the mitochondrial landscape in differentiating cells. Based on these first findings, we intend now to investigate the mitochondrial metabolic function changes occurring in differentiating 3T3-L1 exposed to either LG-GAL ((Low glucose (5 mM glucose) + galactose (20 mM galactose)) or HG (25 mM glucose) by using targeted ^1^H-NMR (proton nuclear magnetic resonance) spectroscopy of cellular fluids (intracellular extract and extracellular media). Hence, selected mitochondrial metabolic markers were relatively scored from their spectral intensities to perform the statistical comparison of their abundances in both carbohydrate conditions all along the adipogenesis process.

## 2. Materials and Methods

### 2.1. Cell Culture and Differentiation

The murine 3T3-L1 cell-line was purchased from the American Type Culture Collection (ATCC) and subcultured in growth media consisting of Dulbecco’s modified Eagle’s medium (DMEM) HG (25 mM glucose, Sigma-Aldrich, St. Louis, MO, USA) supplemented with 10% fetal bovine serum (FBS) premium (Pan-Biotech, Aidenbach, Germany). At passage 7 (P7), cells were cultured until reaching 100% confluence and then let for two additional days before starting the differentiation (D0). At D0, cells were exposed for 3 days to a differentiation medium (DM) consisting of culture medium + 10% FBS + an adipogenic cocktail (insulin 1 mg/mL (Sigma-Aldrich, St. Louis, MO, USA), IBMX 12 mg/mL (Sigma-Aldrich, St. Louis, MO, USA) and DEX 0.4 mg/mL (Sigma-Aldrich, St. Louis, MO, USA)). The culture medium was different depending on the condition: DMEM HG (25 mM glucose, Sigma-Aldrich St. Louis, MO, USA) was used for the HG condition and a DMEM LG (5 mM glucose, Sigma-Aldrich, St. Louis, MO, USA) + GAL (20 mM galactose, Sigma-Aldrich, St. Louis, MO, USA) for the LG-GAL condition. At D3, the DM was replaced by a maintenance medium (MM) containing the same components as DM except for IBMX and DEX. MM was replaced every two days until D7 that corresponds to mature adipocytes. Cells were harvested on D0, D3, D5 and D7 to perform intracellular metabolites extraction steps. Medium of cells was sampled for extracellular metabolites levels assessment before being changed or renewed.

### 2.2. Extracellular (EC) and Intracellular (IC) Metabolic Extracts Preparation and 1H-NMR Spectra Acquirement

EC and IC metabolites levels were determined on D0, D3 and D7 of differentiating 3T3-L1 cells cultured either in LG-GAL or HG medium. EC metabolites levels were assessed on culture media, and IC metabolites levels were measured after chloroform–methanol–water extraction separating the cell content into two main phases (one hydrophilic and one hydrophobic). Briefly, cells were washed twice with warmed DPBS, quenched with cold water:methanol solution (1:4) and successively vortexed and sonicated following the addition of the different solvents (methanol, water and chloroform; (final ratio: 1:0.5:0.5)). Following this mechano-chemical extraction phase, the extract solution was centrifuged 10 min at 14,000× *g* to obtain the different phases. The upper phase (methanol–water phase) containing hydrophilic metabolites was collected, evaporated with a speed vacuum, and resuspended using 700 µL phosphate buffer (80:20 D2O), pH 7.4. Extracellular media samples (500 µL) were mixed with a phosphate-buffered (80:20 D2O) solution (250 µL). IC and EC samples were centrifuged at 10,000× *g* for 10 min. The supernatant was collected (650 µL), mixed with trimethylsilylpropionic acid (TSP) at a final concentration of 1 mM and transferred into 5 mm diameter NMR tubes. 1H-NMR spectra of the intracellular aqueous phase and extracellular fluids were acquired using a Bruker Avance 600 MHz spectrometer with a 5 mm PABBO BB-probe. A NOESYPRESAT-1D sequence was used with 256 scans. The acquired FIDs were Fourier-transformed to obtain spectra. The program Mestre Nova 11 (Mestrelab Research, Santiago de Compostela, Spain) was used for further spectral processing (baseline, reference and phase corrections).

### 2.3. Analytical Steps of the IC and EC Metabolic Contents through a Two-Step 1H-NMR-Based Metabonomic Study

This study included two successive analytical steps: (1) preliminary exploratory untargeted metabonomic study and (2) selection and semiquantitative assessment of key metabolites.

## 3. Untargeted Exploratory Approach

### 3.1. Multivariate Date Analyses

The first step of this metabonomic study was based on multivariate data analyses of the NMR spectral data set in terms of observations (biological samples) and variables (metabolites) discrimination between the two carbohydrate conditions (HG vs. LG-GAL). First, NMR spectra were handled using Mestre Nova 11 software. The spectral area from 0.08 to 10 ppm was subdivided into subregions of 0.04 ppm width, and each spectral subregion (descriptor) was integrated. After removal of the water (from 4.2 to 5.1 ppm) and TSP resonances (from 0.08 to 0.8 ppm), each descriptor integral was normalized to the total spectral area. Finally, the numerical values were gathered in an Excel table and exported to SIMCA P + 12 (Umetrics^®^) software to perform multivariate data analysis (statistical projection of the data set). On those projections (centered method), the two main components (1 and 2) were used to evaluate the most discriminant descriptors and, after identification based on their chemical shift and multiplicity, the corresponding metabolites (as shown on the loadings plot in Figure 1A mostly affected in 3T3-L1 cells exposed to LG-GAL medium (in red on the scores plot on Figure 1A) versus HG medium (in black on the scores plot on Figure 1A). Those changes were followed all along the adipogenesis (D0, D3 and D5).

### 3.2. Identification of Discriminant Variables (Metabolites)

Metabolites of interest (corresponding to the discriminant descriptors) were identified from several databases, including the Human Metabolome Database (HMDB), as well as by using the Chenomx Profiler software 8.3 (Chenomx, Edmonton, AB, Canada) (Figure 1B). This part was performed without selecting specific markers, and all the metabolites were considered by performing non-guided analyses of the overall data. Finally, key intracellular metabolic markers (Figure 1C) were listed and used in the second part of this study consisting of selecting the most relevant mitochondrial metabolic markers.

### 3.3. Relative Levels of the Mitochondrial Metabolic Markers’ Assessment

The metabolites selected in the first step were submitted to a relative quantitation based on the AUC (Area Under the Curve) measurement of their best-resolved resonance. For example, glucose was assessed from its doublet resonance at 5.25 ppm. Then, the data were expressed as a percentage of the corresponding value obtained on Day 0 under the HG condition. For instance, in the case of glucose, the mean AUC of the doublet resonance measured on D3 in LG-GAL condition was divided by its mean AUC measured on D0 in HG condition. Such a relative assessment of the metabolic levels allowed us to statistically express some metabolic differences between the two conditions.

For a particular metabolite of interest, the relative expression of its levels was based on the following formulation (Equation (1))
(1)$$\frac{Mean\;AUC\;of\;best\;resolved\;resonance\;at\;\left(D0\;or\;D3\;or\;D7\right)\;in\;\left(LG-GAL\;or\;HG\right)}{Mean\;AUC\;of\;best\;resolved\;resonance\;at\;D0\;in\;HG\;condition}$$

Relative metabolites of interest levels calculation

The AUC integration was performed using a “peak picking tool” available in Mestre Nova 11 software (Mestrelab Research, Santiago de Compostela, Spain).

This (Table 1) semiquantitative method was validated by assessing the relative glucose levels in our LG (5 mM glucose) vs. HG media (25 mM glucose). A ratio of ca 1:5 was calculated, as expected from the glucose levels in LG versus HG conditions.

### 3.4. Selection of Mitochondrial Metabolic Markers

Once the relative levels of the metabolites were determined, metabolites (underlined and in bold in Figure 1C) were considered as statistically relevant and somehow related to the mitochondrial metabolism (catabolism vs. anabolism, anaplerotic reactions, oxidative vs. anaerobic metabolism) were selected. Those selected intracellular markers were also checked and semiquantitatively assessed in the extracellular media of differentiating 3T3-L1 cells exposed to either HG or LG-GAL.

## 4. Targeted Approach

After selecting mitochondrial metabolites of interest, a targeted ^1^H-NMR-based metabonomic study of the mitochondrial metabolome was performed. Based on histograms and statistical analyses of the relative metabolite levels, a semiquantitative investigation of the different related mitochondrial metabolic pathways was performed. Different aspects of the mitochondrial metabolome were assessed through selecting those markers and are detailed in the discussion.

### Statistical Analyses

Data from at least three independent experiments (n = 3, biological triplicate) were analyzed using two-way ANOVA and Holm–Sidak’s multiple-comparison test. Statistical analyses were performed using GraphPad Prism 6 software. Results are presented as mean values ± SEM. The level for statistical significance was defined as *p* < 0.05.

## 5. Results

### 5.1. Selection of Mitochondrial Metabolic Markers

#### Carbohydrates Supply Intake and Consumption

Exposing differentiating 3T3-L1 cells to either 25 mM glucose or to 5 mM glucose + 20 mM galactose was expected to give different EC (Extracellular) and IC (Intracellular) metabolic profiles from cells harvested at the different time-points and from the two compared conditions [18]. As expected, EC and IC glucose levels (Figure 2A,B) were considerably lower in the LG-GAL condition, whose exclusive EC and IC galactose (Figure 2C,D) availability showed a different carbohydrate profile. Regarding IC galactose in differentiating 3T3-L1cells, part of it was converted into galactitol. Galactose conversion into glycolytic subproduct (Glucose-1-phosphate further isomerized into glucose-6-phosphate) occurring in the liver is represented using a dotted arrow in Figure 2. In the WAT, the other pathways using galactose as a substrate were not detected and are also represented by a dotted arrow ending by a question mark.

### 5.2. Carbohydrate Metabolism-Anaerobic Glycolysis

Evaluating both glucose and lactate as indicators of anaerobic glycolysis was performed to better investigate the downstream catabolic changes associated with glucose and partial galactose replacement (Figure 3). In the LG-GAL condition, the decrease of glucose in both the EC and IC media (Figure 2) indicates the progressive restriction in glucose availability and increased glucose transport and consumption over adipogenesis (D0-D7). In addition, inducing adipogenesis in lower glucose conditions (LG-GAL), compared to HG conditions, led to significantly lower IC and EC lactate levels (D7 ****), indicating lower production and secretion of this anaerobic glycolysis rate marker.

### 5.3. Carbohydrate and Related Mitochondrial Catabolism—TCA (TriCarboxylic Acid) Cycle

To continue the carbohydrate metabolism assessment of 3T3-L1 cells differentiating in either HG or LG-GAL conditions, a closer look at the mitochondrial metabolism was performed (Figure 4). Closely linked to their function of the “powerhouse” of the cells, the mitochondrial TCA (TriCarboxylic Acid) cycle is associated with both carbohydrate catabolism and cellular energy production [19,20]. Several key markers of the TCA cycle were semiquantitatively assessed, giving an overview of the TCA cycle activity in both conditions. Three main TCA cycle markers were relatively quantified in the IC media of 3T3-L1, and two of them were also detected and evaluated in the EC media. Our results pinpoint lower levels of the selected TCA markers in the intracellular fraction with statistically significant differences reached at D7 for all of them (pyruvate (**), succinate (**) and fumarate (****)) in LG-GAL condition (Figure 4). Those highly significant differences were correlated to two opposite trends observed in TCA cycle activity through adipogenesis depending on the carbohydrate supply. Whereas in HG condition, TCA markers levels become significantly higher at D7, levels of those metabolites constantly decrease over the adipogenesis in LG-GAL condition. Pyruvate, initially present at similar levels in both HG and LG DMEM culture media (Figure 4A), appears to be slightly lower in LG-GAL condition than in HG condition. In this last condition, EC pyruvate levels were kept constant over adipogenesis. On the other hand, in the LG-GAL condition, EC media displayed lower levels reflecting lower IC pyruvate levels (Figure 4C) due to lower consumption and/or secretion. As for succinate (Figure 4B), whereas its EC levels tended to increase in HG condition, those levels decreased in LG-GAL condition through adipogenesis. Absent from both HG and LG DMEM culture media, succinate levels modulation could only be attributed to altered secretion.

### 5.4. Mitochondrial Metabolism—Non-Glycolytic “Glutamine–Glutamate–Pyroglutamate” Group

Besides mitochondrial metabolism related to carbohydrate combustion, other metabolites could be assessed to evaluate the mitochondrial metabolic function status (Figure 5). The glutamine–glutamate–pyroglutamate group can be considered as non-glycolytic since there are not directly coming from a glycolytic source. In addition, the glutamine oxidation into glutamate further converted into alpha-ketoglutarate is a decisive reaction to sustain TCA cycle activity [21]. For this reason, glutamine and glutamate were considered, in our study, as two markers of mitochondrial metabolic function. Thanks to our previous ^1^H-NMR metabolomics study [18], pyroglutamate was retained as a third informative metabolite. Indeed, less is known about pyroglutamate in mammalian cells if it is only its production from glutamate cyclization [22]. Although different trends were observed inside this glutamine–glutamate–pyroglutamate group, the combined findings could be used to partially reflect the mitochondrial metabolic status by considering glutamate as a central hub. Indeed, IC glutamate levels (Figure 5C) significantly (****) and similarly decreased in both conditions through adipogenesis (D3 and D7). However, EC glutamate levels did not follow the same trend, with a significant increase in the HG condition and a late significant decrease at D7 in the LG-GAL condition (Figure 5D). This could indicate an increase in glutamate consumption and an associated decrease in its extracellular release in the LG-GAL condition. On the opposite, an increase in glutamate release was observed in HG-condition. The changes observed in IC and EC glutamine levels (Figure 5A,B) clearly indicate a significant increase occurring during the late maturation phase of the adipogenesis (D7 ****) in HG condition, whereas those levels look significantly lower in LG-GAL condition (**). Finally, higher IC and EC pyroglutamate levels (Figure 5E,F) were noticed in LG-GAL (****) condition, most likely due to greater production and release during the early adipogenic phase (D3). During the late adipogenic phase, IC pyroglutamate levels were clearly decreased in LG-GAL conditions and associated with significantly high and constant EC pyroglutamate levels. This could be correlated to a reduction of glutamate into pyroglutamate conversion and/or an increase of its release.

### 5.5. Mitochondrial Metabolism Markers—Acetate and β-hydroxybutyrate

To go deeply into the different pathways that could influence mitochondrial metabolism, we decided to look closer at acetate and the β-hydroxybutyrate, two poorly assessed metabolites in the WAT (Figure 6). Both are known to be closely related to the Acetyl-CoA metabolism. As shown in Figure 6, the early adipogenic phase is associated with a high and significant increase of IC and EC acetate (Figure 6A,B) in both conditions (****). However, the increase of IC acetate was significantly greater in LG-GAL condition than in HG one. However, at the end of the late maturation phase, IC and EC acetate levels significantly decreased in both conditions and particularly in the HG condition (*). At D7, EC levels followed a similar decreasing trend in both conditions. IC β-hydroxybutyrate levels significantly decreased in both conditions after the early adipogenic phase (D0 vs. D3; *). However, such an observation was not correlated to changes in its levels in the EC media at the same time. β-hydroxybutyrate, like acetate, was not initially present in our culture media, and changes in its EC levels (Figure 6C) could only be correlated to a different secretion. During the late maturation phase, two opposite trends were observed depending on the condition. Whereas β-hydroxybutyrate IC levels (Figure 6D) continued to decrease in HG condition, significantly greater levels were observed in LG-GAL condition (**). In the EC media, a similar significant increase of β-hydroxybutyrate levels was observed in both HG and LG-GAL conditions (****).

## 6. Discussion

After evidencing mitochondrial dynamics and functional changes in our previous studies [18], we wanted to look deeper at the IC and EC mitochondrial metabolome changes linked to carbohydrate supply changes in an adipogenic context. Such an investigation aims to characterize the mitochondrial functional changes by evaluating different aspects of this organelle metabolic function (anabolism vs. catabolism, oxidative vs. anaerobic metabolism and anaplerotic pathways) and thanks to a targeted ^1^H-NMR-based metabonomic study of the mitochondrial metabolome. Previously to this targeted approach, an exploratory assessment of the 3T3-L1 intracellular metabolome evolution during adipogenesis was performed in both conditions and by using multivariate analytical tools (Figure 1). This preliminary step was essential to determine the most relevant and biologically significant markers of the mitochondrial metabolic function to be addressed in this targeted metabonomic study (Figure 1C). Therefore, this targeted study was structured into different parts assessing metabolites’ relative levels compared to carbohydrate conditions and linking them to a marker function reflecting a specific part of the mitochondrial metabolic function.

First, we evaluated the fluctuations in IC and EC levels of carbohydrates during adipogenesis when 3T3-L1 cells were exposed to either 5 mM glucose + 20 mM galactose) (LG-GAL) or 25 mM glucose (HG). Unsurprisingly, differentiating 3T3-L1 cells in LG-GAL condition had lower EC and IC glucose levels associated with important galactose intake related to this carbohydrate EC availability (Figure 1). According to our results, in LG-GAL condition, galactose appears to be converted into galactitol, an end-product of a biochemical pathway not associated with energy production. In other words, as expected, adipocytes are not enzymatically equipped to convert galactose back to glucose. However, such an important galactose transport and metabolism indicate that 3T3-L1 cells are highly active in terms of “carbohydrates catabolism” when the carbohydrates supply is high. Next, figuring out such greed for carbohydrates, we investigated different aspects of the metabolism directly linked to carbohydrates intake modulation.

During the hyperglycemia period, WAT avidly consumes glucose [23,24] and, for storage purposes, converts it into a lipid form, triacylglycerols [25]. Also, WAT produces high amounts of lactate [26,27], indicating a pro-glycolytic profile that known to be exacerbated in an obesity context [28]. Our findings confirm that differentiating 3T3-L1 has a higher glycolytic rate in HG condition as displayed by higher glucose consumption and lactate production and secretion (Figure 2). Conversely, reducing glucose availability by a partial replacement with galactose led to lower lactate production, indicating a poorer anaerobic glycolysis stimulation to maintain cell metabolism. Prior studies focused on anaerobic glycolysis when studying 3T3-L1 differentiation [29,30]. A study evidenced increasing IC lactate levels in differentiating 3T3-L1 as a management strategy to deal with an excess of glucose [31]. In addition to this high glycolytic rate, it has been proven in vitro *(in 3T3-L1)* that lactate secretion is high in HG conditions. Transposing to the in vivo situation, this would mimic the liver conversion of lactate to promote neoglucogenesis, essential to face potential further fasting periods [32]. Such an LG-GAL condition, if strictly considered as a glucose restriction situation under the in vivo glycolytic point of view, does not stimulate excess glucose management strategy and slow down lactate release usually secreted by the WAT under high glycemia period. Interestingly, those observations highlight that although being an in vitro model, 3T3-L1 cells demonstrate endocrine-like features by keeping organ crosstalk strategies through metabolites secretion modulation in different metabolic contexts. In addition, our findings emphasize previous observations of differentiating 3T3-L1 adipocytes as obligatory anaerobic glycolytic cells. Such an observation is comforted by recent reports suggesting a signaling role of lactate [33,34,35,36] and evidencing it as mandatory actors to promote 3T3-L1 differentiation [30,37]. In such a context, we propose lactate as a marker of high glucose-associated metabolic stress conditions promoting adipocytes differentiation in WAT expansion contexts.

After evidencing such importance of the anaerobic glycolytic metabolism in differentiating and mature adipocytes, we focused on the mitochondrial metabolic status of differentiating 3T3-L1. Previously, we demonstrated an improvement in the mitochondrial network health status in 3T3-L1 differentiating in LG-GAL condition compared to HG one [18]. As an overall key feature, mitochondria were less stressed under this low glucose condition, and, according to the prior discussion point, this could be correlated to a change in the mitochondrial metabolism associated with less deleterious nutritional conditions. Here, we intended to specifically evaluate selected metabolic markers reflecting different aspects of the mitochondrial function using the ^1^H-NMR metabonomic approach.

TCA cycle appeared as the first key mitochondrial metabolic pathway we wanted to have a closer look at (Figure 4) and mainly due to its potential direct relation with the anaerobic glycolysis rate. In addition, assessing this oxidative catabolic pathway is a common feature of most mitochondrial function assessment methods, such as MTT and cellular oximetry tests. Pyruvate, by being the end-product of aerobic glycolysis and the key precursors of acetyl-CoA further feeding the TCA cycle, appears to be an excellent marker of oxidative carbohydrate catabolism. In turn, succinate and fumarate, by playing the reversible role of substrate and product of the reaction catalyzed by the succinate dehydrogenase (TCA cycle enzyme and complex II of the mitochondrial respiratory chain), were more considered as specific markers of the oxidative mitochondrial catabolism. It is well-known that oxidative mitochondrial metabolism gets stimulated when reaching an advanced adipogenic maturation phase in nearly mature adipocytes [38].. However, it is important to underline that those observations were done in HG conditions and could be a consequence of such “excess glucose” conditions. To this point, our LG-GAL condition, by reducing the glycolytic rate and potentially modulating the mitochondrial metabolism, could give rise to new metabolic considerations. It suggests that in LG-GAL condition, the oxidative catabolism is broadly reduced as evidenced by pyruvate, succinate and fumarate IC lower levels. It is also important to note that some TCA cycle intermediates are known to play signaling roles, which can be accordingly tuned. Recent evidence of succination, a post-translational modification linked to fumarate levels, seems to link this mechanism stimulation to glucotoxicity in the WAT tissue, a phenomenon increasing associated with obesity development and linked mitochondrial dysfunction onset [39,40,41,42]. In addition, It is increasingly acknowledged that under stressful metabolic circumstances, mitochondria tend to release TCA intermediates whose levels can considerably increase in the extramitochondrial compartments [43]. For this purpose, pyruvate EC levels were interestingly lower in LG-GAL conditions than in HG ones. Such an observation could be correlated to higher consumption of this substrate and a lower leaking of it from the intracellular to the extracellular compartment. Remarkably, it appears that IC and EC succinate levels, a substrate not initially present in the DMEM media we used, were considerably higher in HG media after cell harvesting. For this reason, one can reasonably propose that in LG-GAL condition, mitochondrial metabolites tend to stay in the intramitochondrial compartment, reflecting a poorer release of TCA intermediates, markers of mitochondrial stress onset. In other words, the high stimulation of the oxidative catabolism seems to be linked to a glucose overload, consequently stimulating the mitochondrial catabolic function as a protective mechanism justifying our previous observation of more catabolically stressed mitochondria in HG condition.

When considering the mitochondrial metabolism, the metabolism related to the glutamine–glutamate group represents another pathway to keep an eye on (Figure 5). Over the past decades, numerous studies evaluated that fluctuations in glutamine–glutamate levels could reflect relevant changes in mitochondrial function [44]. Interestingly, the observation of very similar glutamate levels, clearly decreasing over time in both LG-GAL and HG conditions, underlines a decisive management mechanism of this metabolite during the differentiation process. However, obvious discrepancies in the evolutions of glutamine and pyroglutamate levels clearly indicate different mitochondrial metabolic profiles in both conditions. In such a context, the significant increase of both IC and EC glutamine levels in the HG conditions, associated with the parallel decrease of glutamate, suggests stimulation of the glutamate conversion to glutamine. This could reflect a mitochondrial overload due to the overstimulation by the carbohydrate catabolism, consequently stimulating glutamate reduction into glutamine. Preventively, such a reaction could prevent further mitochondrial feeding through alpha-ketoglutarate production and alleviate additional mitochondrial catabolic function requirements. The significant increase of the glutamine EC levels suggests an extracellular release mechanism avoiding intracellular glutamine excesses. In agreement with this, a very recent publication highlighted that WAT glutamine levels could be inversely correlated to adiposity and could inhibit glycolysis [45]. Our observations of higher EC glutamine EC release during the late maturation phase of the adipogenesis process could be explained by such a signaling role of this metabolite. Finally, the significant increase of glutamate release and the glutamate conversion into pyroglutamate during the late maturation phase strengthens the hypothesis of a progressive mitochondrial overload and glutamate disposal mechanisms setup. In the LG-GAL condition, the quite different glutamate levels management seems to be more correlated to glutamate into pyroglutamate conversion mechanisms. Less is known about pyroglutamate in mammalian cells, except that its production comes from glutamate cyclization [22]. However, our results show that this metabolite is an additional marker helping to better assess glutamate levels evolution, itself influenced by the mitochondrial metabolic function. We fascinatingly found that 3T3-L1 cells display significantly higher IC pyroglutamate levels in the early adipogenic phase and higher EC pyroglutamate levels across adipogenesis in the LG-GAL condition. Recently, a correlation between poorer visceral adiposity and higher pyroglutamate levels in the WAT of physically trained mice was reported [46]. It clearly appears that the progressive glutamate decrease in LG-GAL condition is first due to its conversion into pyroglutamate, an early anti-adipogenic marker. Over time, pyroglutamate is less produced but is greatly released in the extracellular compartment of mature adipocytes. As for other mitochondrial metabolites, this could underline an extracellular signaling function of pyroglutamate to extend anti-adipogenic power, consequently affecting adipogenesis of still undifferentiated peripheric cells in LG-GAL condition. Through all those observations, we clearly identify different “glutamine–pyroglutamate group” profiles in both conditions, mostly reflecting the prevention role of LG-GAL media against mitochondrial metabolic overload and the associated reduction of the adipogenic yield in the 3T3-L1 cell line. On the other side, the mitochondrial overload occurring in HG seems to prevent anaplerotic mitochondrial pathway triggering by avoiding glutamate levels rising through its reduction into glutamine.

To expect reaching the most comprehensive overview of the mitochondrial metabolism evaluation, we also included some markers reflecting the acetyl-CoA metabolism. As previously noted, acetyl-CoA, a mitochondrial metabolite, is at the crossroad of several biochemical pathways, and its levels can deeply influence key metabolic and cellular pathways [47,48]. Commonly known to be produced from pyruvate glycolytic supply, acetyl-CoA can also be linked to acetate. Some authors already associated acetate levels with mitochondrial metabolism, but also the nutrient availability in several cell types [13,49]. Interestingly, we found that acetate levels are very sensitive to the differentiation induction as evidenced by the significant IC and EC increase of its levels in 3T3-L1 in both conditions during the early adipogenic phase. Such an observation could be correlated to the deep signaling network remodeling associated with the adipogenesis setup and requiring, at some key point, acetylation phenomenon. The similar acetate EC increase in both conditions, reflecting an important metabolite release, could reflect this correlation between differentiation induction and an acetate peak. However, the sharper IC acetate increase in LG-GAL conditions during the early adipogenic phase gave rise to questions about acetate sources. Recently, some authors advanced the hypothesis of a de novo production of acetate in case of environmental metabolic changes. Under certain circumstances, such as mitochondrial poorer function, de novo acetate production seems to be stimulated [49]. Assuming that the mitochondrial network is still not fully functional during the early adipogenic phase and that mitochondrial catabolism stimulation can be increased through glucose overload, acetate production from pyruvate can be a possible explanation. At this stage of differentiation, 3T3-L1 cells in LG-GAL condition display a poorer stimulation of the mitochondrial network maturation reducing the requirement of acetate conversion into acetyl-CoA. Acetate, through a secretion mechanism, is also expected to play an extracellular role, as observed by higher EC acetate levels in the late maturation phase of adipogenesis in LG-GAL conditions. In this condition, acetate production and secretion increase reflect lower acetyl-CoA-dependent mitochondrial function associated with a poorer acetyl-CoA-dependent lipogenic process, a phenomenon reflecting an anti-adipogenic effect. Moreover, to a greater extent, high acetate secretion could also reflect the deep cellular and metabolic changes associated with conditions that are not associated with a nutritional excess, informing by this way, the peripheral tissues of this environmental change. Acetate released by the WAT seems to play a signaling function and could also constitute a metabolizable fuel for peripheric tissue. Interestingly, β-hydroxybutyrate, a ketone body known to be produced by the liver to fuel peripheric tissues when blood glucose levels are reduced [50], also appears to be highly produced and secreted by mature adipocytes in LG-GAL condition. In other words, 3T3-L1 clearly seems to reduce mitochondrial acetyl-CoA metabolism in favor of β-hydroxybutyrate production, therefore, reducing the mitochondrial carbohydrates catabolism and the acetyl-CoA-dependent lipogenic process. In this way, β-hydroxybutyrate can be assessed as a metabolite playing an extra-WAT feeding function. Such an observation reinforces the secretory function of differentiating adipocytes modulating their metabolism to promote systemic metabolic homeostasis in conditions where the glucose is not in excess and probably deleterious. In addition to that, it has been recently proved that β-hydroxybutyrate secretion can be associated with a reduction of fibrosis and can facilitate beige adipogenesis [51]. The observation of such phenomena, linked to a healthier adipose tissue function jeopardizing obesity development or progress, underlines the potential systemic benefits of this LG-GAL condition.

## 7. Conclusions

This study aiming at a deeper investigation of the anti-adipogenic and mitochondrial metabolic function modulation in LG-GAL condition gave rise to new considerations about the metabolism assessment thanks to a targeted ^1^H-NMR-based metabolomics approach used.

From a biological point of view, this study confirmed that such a glucose reduction and partial replacement by galactose is associated with anti-adipogenic effect and several mitochondrial metabolic-related pathway changes (as summarized in Figure 7). HG condition appeared as an “excess glucose” condition highly stimulating carbohydrates catabolism (glycolysis and acetyl-CoA related mitochondrial carbohydrates catabolism), further stimulating the adipogenic and storage function of 3T3-L1 cells. The reduction of the glycolytic rate associated with the stimulation of mitochondrial metabolic pathways not associated with the carbohydrates catabolism (higher glutamine into glutamate and subsequent glutamate into pyroglutamate transformation, higher acetate and β-hydroxybutyrate levels, and lower lactate and TCA intermediates levels), in LG-GAL condition, evidence a deep mitochondrial metabolic function changes linked to a reduction of the adipogenic yield and stimulation of the secretion of signaling metabolites (lower lactate and TCA intermediates release associated with higher pyroglutamate, acetate and β-hydroxybutyrate secretion). Consequently, this study throws light on clear but still poorly known metabolite functions as markers of the improvement of the mitochondrial network health status associated with poorer deleterious and less adipogenic nutritional conditions in differentiating 3T3-L1 cells.

Finally, by assessing different aspects of the mitochondrial metabolism (non-exclusively related to the carbohydrates metabolism), this targeted metabonomics study throws light on a new way to evaluate the mitochondrial function, not only dedicated to the mitochondrial oxidative metabolism. Most of the current mitochondrial metabolism assays (MTT, TCA and mitochondrial respiratory assessments) are only focusing on one part of the mitochondrial function and lose a plethora of information to evaluate a potential mitochondrial dysfunction onset. The present study highlights that claiming a mitochondrial dysfunction based on only an oxidative carbohydrates catabolism assessment is irrelevant, and our results should stimulate other mitochondrial markers investigation assessment (acetate, glutamine, glutamate, pyroglutamate, acetate, β-hydroxybutyrate levels) to better reflect the mitochondrial function and health status.

## Figures and Tables

**Figure 1 biomolecules-11-00662-f001:**
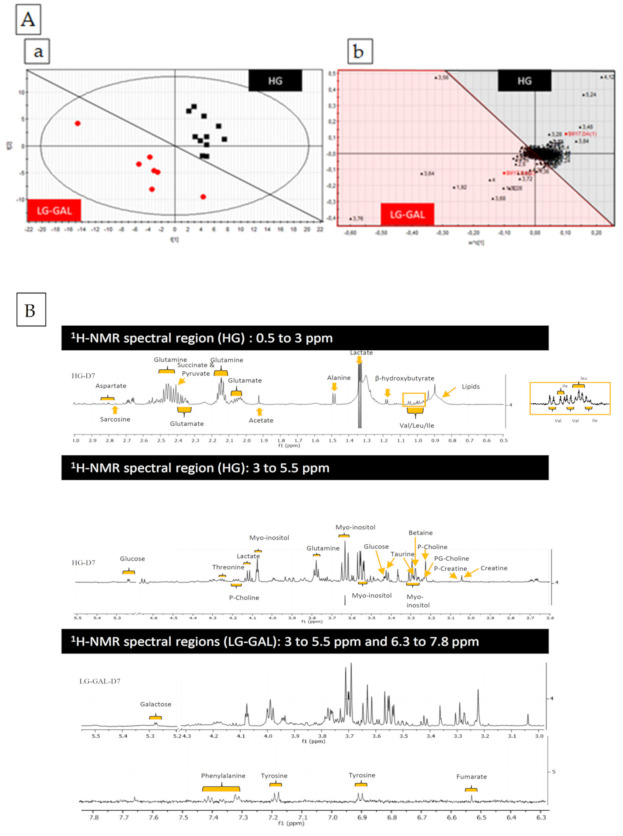
Preliminary and exploratory metabonomic study: mitochondrial markers selection. (**A**) Preliminary multivariate data analyses: (**a**) Scores scatterplot (PLS-DA model) of the intracellular metabolites of 3T3-L1 cells cultured either with LG-GAL or HG media and harvested at the different time-points (D0, D3 and D5). R^2^_cum_ = 0.871; Q^2^_cum_ = 0.525; Hotelling’s T2 = 0.95; *p*-value (CV-ANOVA) = 0.0223. (**b**) Loadings scatterplot (PLS-DA model) of the intracellular metabolites of 3T3-L1 cells cultured either with LG-GAL or HG media and harvested at the different time-points (D0, D3 and D5). (**B**) ^1^ H-NMR intracellular metabolome of 3T3-L1 cells (identified hydrophilic metabolites). (**C**) Identified polar intracellular metabolites based on the previous multivariate analyses of the dataset and on identifying the detected peaks on (**A**): Preliminary and exploratory metabonomic study—mitochondrial markers the ^1^H-NMR spectra of the polar intracellular extract through Chenomx tool^®^. Selected metabolites are in bold and underlined. Statistical (* *p*< 0.05; ** *p* < 0.01; *** *p* < 0.001; **** *p* < 0.0001) comparison between LG-GAL vs. HG were performed by two-way ANOVA and Holm–Sidak’s multiple-comparison.

**Figure 2 biomolecules-11-00662-f002:**
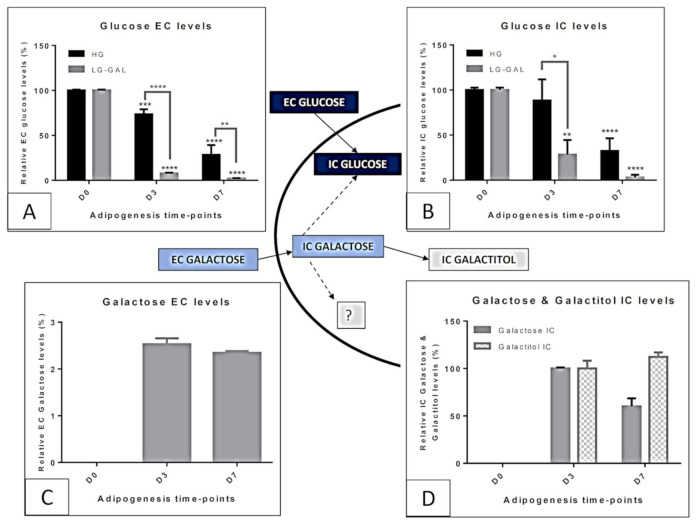
Evolution of IC (Intracellular) and EC (Extracellular) levels of key carbohydrates through adipogenesis of 3T3-L1 cells in HG vs. LG-Gal conditions. Dotted arrows mean uncertain/hypothetical galactose conversions. (**A**) Relative glucose EC levels. (**B**) Relative glucose IC levels. (**C**) Relative galactose EC levels. (**D**) Relative galactose and galactitol IC levels. Results are the means ± SEM (biological (n = 3) triplicates). Statistical (n.s, * *p* < 0.05; ** *p* < 0.01; *** *p* < 0.001; **** *p* < 0.0001) comparison of values obtained at each key time points and compared to D0 (in black) and between HG vs. LG-GAL (in gray) were performed by two-way ANOVA and Holm–Sidak’s multiple-comparison test.

**Figure 3 biomolecules-11-00662-f003:**
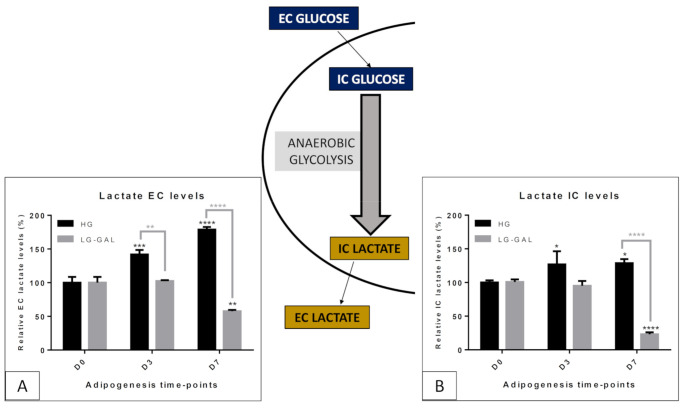
Key anaerobic glycolysis-related metabolites IC (Intracellular) and EC (Extracellular) levels evolution through adipogenesis in both HG vs. LG-Gal conditions scheme. Each graph individually represents either the relative IC or the EC metabolite levels evolution in 3T3-L1 cells. (**A**) Relative lactate EC levels. (**B**) Relative lactate IC levels. Results are the means ± SEM (biological (n = 3) triplicates). Statistical (n.s, * *p* < 0.05; ** *p* < 0.01; *** *p* < 0.001; **** *p* < 0.0001) comparison of each key time points to the D0 situation (in black) and between HG vs. LG-GAL (in gray) were performed by two-way ANOVA and Holm–Sidak’s multiple-comparison test.

**Figure 4 biomolecules-11-00662-f004:**
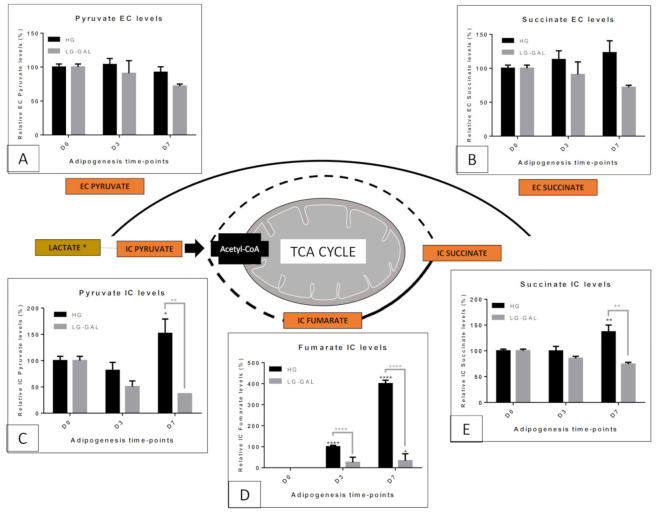
Key oxidative carbohydrates catabolism and TCA cycle-related metabolites IC and EC levels evolution through adipogenesis in both HG vs. LG-Gal conditions scheme. Each graph individually represents either the relative IC or the EC metabolite levels evolution in 3T3-L1 cells. (**A**) Relative pyruvate EC levels. (**B**) Relative succinate EC levels. (**C**) Relative pyruvate IC levels. (**D**) Relative fumarate IC levels. (**E**) Relative succinate IC levels. Results are the means ± SEM (biological (n = 3) triplicates). Statistical (n.s, * *p* < 0.05 ; ** *p* < 0.01 ; **** *p* < 0.0001) comparison of each key time points to the D0 situation (in black) and between HG vs. LG-GAL (in gray) were performed by two-way ANOVA and Holm–Sidak’s multiple-comparison test.

**Figure 5 biomolecules-11-00662-f005:**
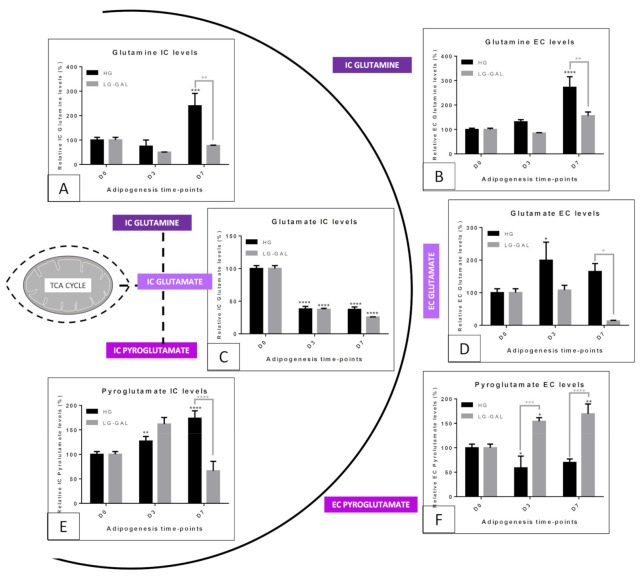
The glutamine, glutamate and pyroglutamate group IC and EC levels evolution through adipogenesis in both HG vs. LG-Gal conditions scheme. Each graph individually represents either the relative IC or the EC metabolite levels evolution in 3T3-L1 cells. (**A**) Relative glutamine IC levels. (**B**) Relative glutamine EC levels. (**C**) Relative glutamate IC levels. (**D**) Relative glutamate EC levels. (**E**) Relative pyroglutamate IC levels. (**F**) Relative pyroglutamate EC levels. Results are the means ± SEM (biological (n = 3) triplicates). Statistical (n.s, * *p* < 0.05; ** *p* < 0.01; *** *p* < 0.001; **** *p* < 0.0001) comparison of each key time points to the D0 situation (in black) and between HG vs. LG-GAL (in gray) were performed by two-way ANOVA and Holm–Sidak’s multiple-comparison test.

**Figure 6 biomolecules-11-00662-f006:**
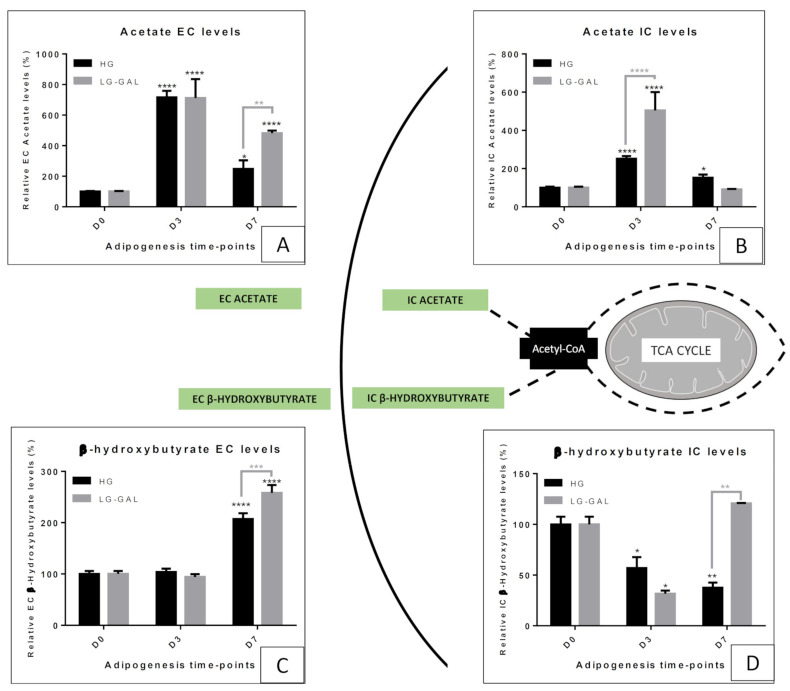
Acetate-β-hydroxybutyrate IC and EC levels evolution through adipogenesis in both HG vs. LG-Gal conditions scheme. Each graph individually represents either the relative IC or the EC metabolite levels evolution in 3T3-L1 cells. (**A**) Relative acetate EC levels. (**B**) Relative acetate IC levels. (**C**) Relative β-hydroxybutyrate EC levels. (**D**) Relative β-hydroxybutyrate IC levels. Results are the means ± SEM (biological (n = 3) triplicates). Statistical (n.s, * *p* < 0.05; ** *p* < 0.01; *** *p* < 0.001; **** *p* < 0.0001) comparison of each key time points to the D0 situation (in black) and between HG vs. LG-GAL (in gray) were performed by two-way ANOVA and Holm–Sidak’s multiple-comparison test.

**Figure 7 biomolecules-11-00662-f007:**
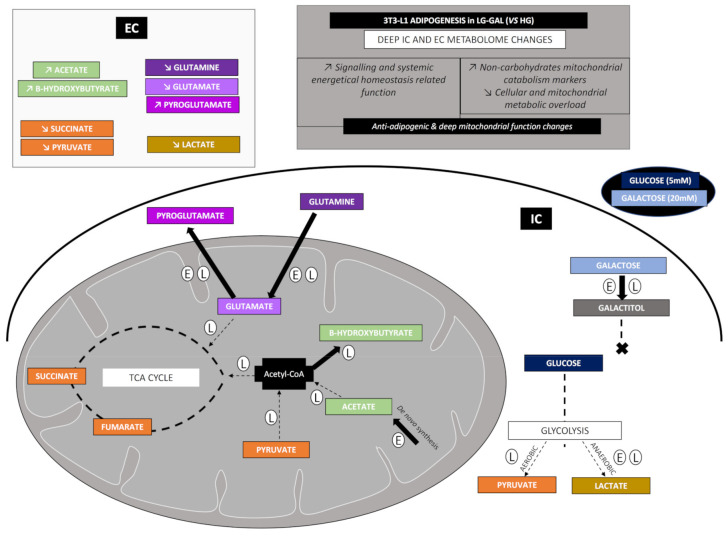
Summary of the main metabolic markers’ changes in 3T3-L1 differentiating in LG-GAL condition compared to the HG condition. E and L surrounded letters stands, respectively, for early and late adipogenic phases. The filled arrow stands for a pathway that is higher in the LG-GAL condition compared to the HG condition. A discontinued arrow stands for a pathway that is lower in LG-GAL condition compared to the HG. Condition. IC and EC stand, respectively, for intracellular and extracellular media.

**Table 1 biomolecules-11-00662-t001:** Validation of the relative metabolite levels assessment. Integrated AUC values were determined using the peak-picking tool, and relative values were determined based on the previously explained relative levels formulation.

	Total AUC of Doublet at 5.25 ppm	% (Relative to the Mean of the Glucose AUC in HG Media)
Glucose EC levels in HG media	4.87	100%
Glucose EC levels LG media	0.97	19.9%

## Data Availability

The authors confirm that the data supporting the findings of this study are available within the present article.

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
