# Peer review of "Investigation of Mitochondrial Adaptations to Modulation of Carbohydrate Supply during Adipogenesis of 3T3-L1 Cells by Targeted 1H-NMR Spectroscopy"

_biomolecules, 2021, doi:10.3390/biom11050662_

Round 1
Reviewer 1 Report
This work is the second part of the study published in Nutrients in 2020 with similar conclusion that mitochondria are important in the process of differentiation. The difference in that in the current study the under variations of glucose and galactose metabolites were evaluated using NMR technique what is good enough not being criticized for absence of novelty. The study again is in line with the idea that carbohydrates used can determine mitochondrial metabolism
Major comments
Fig.2 stayed unclear to me. First, I do not understand why the authors repeat the same figures in Fig2 A and B and Fig3A and B. Second, I did not understand how did the authors identify the formation of galactitol. Thirdly, the scheme of galactose metabolism shows the possibly direct formation of glucose from galactose which is novel to me. Fourthly, conventionally galactose can enter glycolysis but not through direct conversion of galactose to glucose rather than through conversion of galactose into glucose-6 phosphate in four steps with participation of glucose 1 phosphate and UDP glucose, and I was surprised that there was no mentioning about this very important and may be the major pathway of galactose metabolism. I have a feeling that namely this four steps chain becomes a rate-limiting in metabolism of adipocytes ultimately causing metabolic starvation. When I analyzed the authors data published in Nutrients 2020, in spite of some apparent defects of that work (although I agreed with the conclusion) I suggested that in some features the authors’ approach to switch to galactose resembles caloric restrictions situation and I would be glad to have some comments on such suggestion.
Fig.3 also raises some questions. Lactate detected in extracellular medium is often explained by the release from dead cells (other metabolites also can be detected) and to exclude this, it is necessary to present dynamics of a cell death in culture provided it takes place during differentiation.
Very strong rise of acetate on D3 is very interesting and I wonder if it correlates with higher non enzymatic protein acetylation and/or being a result of high activity of deacetylases.
Author Response
Please see the attachment, thank you.

Reviewer 2 Report
In the submitted manuscript ID 1173058, Delcourt et al. investigated the glucose and galactose metabolism in white adipose cells (3T3-L1 cell line), using NMR spectroscopic analysis. The data reported in manuscript has highlighted that different supplied carbohydrates (high glucose and low glucose plus galactose) changed the adipose cell metabolism during differentiation.
The manuscript is written in an unclear way and same data are reported confusionally.
In detail, in figure 2A and 2B are showed the same reported in Figure 3A and 3B, respectivelly. Futhermore, in Figure 3A an incorrect abscissa name was reported.
Main issues
In the text the corrisponding number of the described figures should be indicated
Figure 2C: Why galactose EC levels are equal to zero at time point D0?
Figure 3: the data of box A and B are the same of box A and B of Figure 2. For this reasons, in figure 3 only the data about lactate IC levels should be reported.
Lane 258-259: The authors should check if the affermation is relative to EC succinate levels or EC fumarate levels
For a better readability the metabolic schemes in the figures should be coloured, especially for the Figure 7
Minor issues
Figure 1.2 should be renamed as Figure 1.C.
Figure 1.3 should be reported as Table 2
Author Response
Please, see the attachment, thank you.

Round 2
Reviewer 1 Report
I think that the authors properly addresses my critique and I can give a green light for publication
Reviewer 2 Report
The authors have addressed all the concerns, therefore, the manuscript is now acceptable for publication.